# Ramen Consumption and Gut Microbiota Diversity in Japanese Women: Cross-Sectional Data from the NEXIS Cohort Study

**DOI:** 10.3390/microorganisms11081892

**Published:** 2023-07-26

**Authors:** Jonguk Park, Hiroto Bushita, Ayatake Nakano, Ai Hara, Hiroshi M. Ueno, Naoki Ozato, Koji Hosomi, Hitoshi Kawashima, Yi-An Chen, Attayeb Mohsen, Harumi Ohno, Kana Konishi, Kumpei Tanisawa, Hinako Nanri, Haruka Murakami, Motohiko Miyachi, Jun Kunisawa, Kenji Mizuguchi, Michihiro Araki

**Affiliations:** 1Artificial Intelligence Center for Health and Biomedical Research, National Institutes of Biomedical Innovation, Health and Nutrition, 3-17 Senrioka-shinmachi, Settsu 566-0002, Osaka, Japan; 2International Life Sciences Institute (ILSI) Japan, Gobel Building 3-13-5, Morishita, Koto 135-0004, Tokyo, Japan; 3Health & Wellness Products Research Laboratories, Kao Corporation, Tokyo 131-8501, Japan; 4Milk Science Research Institute, Megmilk Snow Brand Co., Ltd., 1-1-2 Minamidai, Kawagoe 350-1165, Saitama, Japan; 5Future Design Division, The KAITEKI Institute, Inc., Palace Building 1-1, Marunouchi 1-chome, Chiyoda 100-8251, Tokyo, Japan; 6Laboratory of Vaccine Materials and Laboratory of Gut Environmental System, Microbial Research Center for Health and Medicine, National Institutes of Biomedical Innovation, Health and Nutrition, 7-6-8 Saito-Asagi, Ibaraki 567-0085, Osaka, Japan; 7Department of Physical Activity Research, National Institutes of Biomedical Innovation, Health and Nutrition, 3-17 Senrioka-shinmachi, Settsu 566-0002, Osaka, Japan; 8Department of Nutrition, Kiryu University, 606-7 Azami, Kasakake-machi, Midori 379-2392, Gunma, Japan; 9Faculty of Food and Nutritional Sciences, Toyo University, 1-1-1 Izumino, Itakura, Oura 374-0193, Gunma, Japan; 10Faculty of Sport Sciences, Waseda University, 2-579-15 Mikajima, Tokorozawa 359-1192, Saitama, Japan; 11Faculty of Sport and Health Science, Ritsumeikan University, 1-1-1 Nojihigashi, Kusatsu 525-8577, Shiga, Japan; 12International Research and Development Center for Mucosal Vaccines, Institute of Medical Science, University of Tokyo, 4-6-1 Shirokanedai, Minato 108-8639, Tokyo, Japan; 13Graduate School of Medicine, Graduate School of Pharmaceutical Sciences, Graduate School of Dentistry, Graduate School of Sciences, Osaka University, 1-1 Yamadaoka, Suita 565-0871, Osaka, Japan; 14Department of Microbiology and Immunology, Graduate School of Medicine, Kobe University, 7-5-1 Kusunoki, Chuo, Kobe 650-0017, Hyogo, Japan; 15Research Organization for Nano and Life Innovation, Waseda University, 513 Waseda-Tsurumaki, Shinjuku 162-0041, Tokyo, Japan; 16Institute for Protein Research, Osaka University, 3-2 Yamadaoka, Suita 565-0871, Osaka, Japan; 17Graduate School of Medicine, Kyoto University, 54 Shogoin-Kawahara-cho, Sakyo-ku, Kyoto 606-8507, Japan; 18Graduate School of Science, Technology and Innovation, Kobe University, 1-1 Rokkodai, Nada-ku, Kobe 657-8501, Hyogo, Japan; 19National Cerebral and Cardiovascular Center, 6-1 Kishibe-Shinmachi, Suita 564-8565, Osaka, Japan

**Keywords:** diet, gut microbiota, ramen, alpha diversity, nutrients

## Abstract

A cross-sectional study involving 224 healthy Japanese adult females explored the relationship between ramen intake, gut microbiota diversity, and blood biochemistry. Using a stepwise regression model, ramen intake was inversely associated with gut microbiome alpha diversity after adjusting for related factors, including diets, Age, BMI, and stool habits (β = −0.018; r = −0.15 for Shannon index). The intake group of ramen was inversely associated with dietary nutrients and dietary fiber compared with the no-intake group of ramen. Sugar intake, *Dorea* as a short-chain fatty acid (SCFA)-producing gut microbiota, and γ-glutamyl transferase as a liver function marker were directly associated with ramen intake after adjustment for related factors including diets, gut microbiota, and blood chemistry using a stepwise logistic regression model, whereas *Dorea* is inconsistently less abundant in the ramen group. In conclusion, the increased ramen was associated with decreased gut bacterial diversity accompanying a perturbation of *Dorea* through the dietary nutrients, gut microbiota, and blood chemistry, while the methodological limitations existed in a cross-sectional study. People with frequent ramen eating habits need to take measures to consume various nutrients to maintain and improve their health, and dietary management can be applied to the dietary feature in ramen consumption.

## 1. Introduction

Food intake is well known to be one of the main factors affecting conventional biomarkers in gut microbiota and blood biochemistry, including blood pressure. In fact, the intake of fruits, vegetables, and dietary fiber is known to help prevent cancer and cardiovascular disease and reduce the risk of death [1,2,3]. A balanced and diverse diet is widely known to be beneficial to health. However, in general, individuals can determine what, when, and how much to eat without much effort, and the intake of nutrients through food intake varies greatly from individual to individual [4]. Therefore, understanding and evaluating the effect of food intake on gut microbiota and blood biochemistry remains a crucial task in the context of human health and longevity research, such as non-communicable diseases.

Meanwhile, a recent international comparison of mortality statistics confirmed that Japan has a significantly lower non-communicable disease mortality rate and the longest life expectancy among the G7 countries, particularly among women [5]. The low mortality rate in Japan compared to other countries seems to reflect the low obesity rate, and the low obesity rate is largely due to Japan’s distinctive dietary pattern, as characterized by low intake of red meat, high intakes of fish, plant foods, and non-sugar sweetened beverages [5]. Studies involving cohort– or case–control-based epidemiological research designs that define dietary patterns of Japanese diets have found that Japanese diets are diverse and are characterized by foods such as soybeans, seafood, and vegetables [6]. In addition, a cohort study of only Japanese individuals confirmed that the intake of a Japanese diet was negatively related to the risk of developing cardiovascular disease and death [7]. Therefore, using a Japanese female cohort with a more diverse diet and a range of health issues can help in understanding the relationship between food intake and health.

Recently, there has been an interest in a healthy intestinal environment as well as eating habits for a healthy life. Recent studies have demonstrated the role of the intestinal environment in human health, which is an active research topic [8]. Along with eating habits, gut bacterial communities are one of the most influential factors in the intestinal environment and are directly or indirectly associated with diseases such as intestinal tract, metabolic syndrome, and neurological diseases by dysbiosis or certain gut bacteria [9,10,11]. In addition, diet and gut bacteria are closely related because diet plays a role in determining the composition of gut bacteria and the number of metabolites in the intestinal environment [12,13,14]. Therefore, understanding the relationship between eating habits and gut bacterial community structure, as well as improving eating habits, are important because they can affect human health and longevity. For example, alpha diversity in gut microbiota affects human health, with low alpha diversity associated with detrimental effects on host health [15], while some dietary patterns, including high vegetables, are directed to higher microbial diversity [16]. Lifestyle and clinical factors correlate with Shannon diversity, such as biomarkers for diabetes, inflammation, liver function, and cholesterol [16]. Additionally, n-3 polyunsaturated fatty acids and other markers related to fish intakes, such as docosahexaenoic acid and mercury, are directly correlated with diversity. Diet and physical activity are also correlated with microbiome diversity. From the viewpoint of clinical intervention on gut microbiota, fecal microbiota transplantation emerged as a promising option for treating *Clostridium difficile* infection [17].

A diet comprising fish and n-3 polyunsaturated fatty acids, seaweed, soybean products, and green tea is typically found in washoku, a traditional Japanese diet [18]. The Japanese diet partly explains the world’s highest life expectancy from a dietary perspective. Although some Japanese cohort studies have been used to interpret the relationship between eating habits and gut bacteria [19,20,21,22], very few have investigated the relationship between overall eating habits and gut bacterial community structure. Moreover, recent cohort studies have shown that sex affects the gut bacterial community [20,23], implying that gender consideration studies are needed for more accurate evaluation, but this is more limited [24]. Yoshikata et al. analyzed the relationship between intestinal bacterial diversity, dietary intake patterns, and urinary equol concentrations in postmenopausal Japanese women but not in blood data, which can be used as a health indicator.

We thought that dietary habits could influence the highly personalized gut bacterial community and could affect human health complexly with gut bacteria, although other factors could affect the gut bacteria, such as genetic and anthropometric backgrounds, biological aging, and infections. To make this relationship clear, we investigated the relationship between habitual diet intake, gut microbial diversity, and blood biochemistry in 224 healthy Japanese adult females aged 27–80 years in a cross-sectional study. We first analyzed the correlation between eating habits and the alpha diversity of gut bacteria, and as a result, we confirmed that ramen intake was most strongly related to the alpha diversity of gut bacteria. The ramen covered in this study refers to the Japanese-style ramen different from the traditional Chinese ones. Various kinds of ramen exist and are generally served in a broth flavored with soy sauce and miso, with typical toppings such as sliced pork, nori, menma, and scallions. A reduction in dietary sodium intake is a cost-effective public health approach to reduce non-communicable diseases, and noodles, especially instant noodles, are an ultra-processed food and typically contain high sodium from a diet [25]. The relationship between ramen and gut bacterial diversity confirmed in our study was consistent with the results of a previous study of Japanese women. Previous studies have reported that increased food intake from meat, fish, beans, vegetables, and Japanese snacks has a positive correlation with microbial diversity, while high consumption of ramen and smoking have a negative correlation [24]. However, whether the relationship between ramen and microbial diversity is limitedly observed in postmenopausal women is still unclear. Moreover, in this study, we set up groups based on ramen intake and evaluated the possible effects on nutrient intake, intestinal bacteria, and human health through a comparative analysis between groups. The results of this study will provide a feature of diet associated with gut microbiota and blood biochemistry, including blood pressure.

## 2. Materials and Methods

### 2.1. Participants

Healthy Japanese adults were included in the source cohort, named the Nutritional and Physical Activity Intervention Study (NEXIS) cohort study since 2012, from October 2015 to June 2019 (ethical approval number: KENEI102; clinical trial registration number: NCT00926744) [26]. Then, some participants with fecal samples were assigned to the additional analysis for the gut microbiome in this study and the related study about fecal microbiome analyses. In this cross-sectional study, human fecal samples were collected from 224 healthy adult women (aged 27–80 years) living in Tokyo metropolitan area, Japan [20,22] (Table 1). The participants lived a healthy lifestyle and were followed up regularly at an institute of the study team, the National Institutes of Biomedical Innovation, Health, and Nutrition (NIBIOHN). The exclusion criteria for the source cohort were the presence of any history of cancer, cardiovascular disease, liver disease, or gastrointestinal disease and the participants who took antibiotics, laxatives, and antiflatulents within a month at the enrolment. In addition, the subjects whose estimated total energy intake was under 600 kcal or over 4000 kcal were excluded due to a lack of data reliability. Written informed consent was obtained from all the participants. This study was approved by the Ethical Committee of NIBIOHN (approval no. KENEI3-07).

### 2.2. Food Frequency Questionnaire

Considering the nature of the Japanese diet comprising local seafood, vegetables, and beverages, dietary habits were assessed using a mail-in, brief self-administered diet history questionnaire (BDHQ) [27]. The amount of nutrient intake was calculated using dietary habit information during the last month for answering the questionnaire. The BDHQ is a short version of a self-administered diet history questionnaire that was developed in Japan and asks about the frequency of consumption in the food list based on the National Health and Nutrition survey. The BDHQ is a four-page fixed-portion questionnaire that asks about the intake frequency of the selected food items to estimate the dietary intake of 58 food and beverage items in the preceding month. The BDHQ consists of 5 sections: (1) intake frequency of food and nonalcoholic beverage items generally found in the Japanese diet, (2) daily intake of rice and miso soup as a common combination of staple food and soup in the Japanese diet, (3) frequency of drinking and amount per drink for alcoholic beverages, (4) usual cooking methods in line with Japanese cuisine, and (5) general dietary behavior [28]. A face-to-face interview was optionally conducted with the participants to review and correct the missing and inappropriate answers. We adopted the energy-adjusted intakes for foods, energy, and nutrients (per 1000 kcal and day) using an ad hoc computer algorithm for further analyses based on the BDHQ validation study against 16-day dietary records in 4 persons [27,28]. The list of food items and nutrients were enumerated in the first 50 rows or the rest rows of Dietary Habit in Appendix A, respectively.

### 2.3. Fecal Sample Collection

A kit for fecal collection and storage was mailed to the participants. Fecal samples were collected from the 244 individuals and stored at 4 °C in guanidine thiocyanate solution (TechnoSuruga Laboratory, Shizuoka, Japan) until DNA was extracted for 16S rRNA gene amplicon sequencing. Another set of fecal samples was stored at −80 °C for short-chain fatty acid (SCFA) measurements. Also, according to our previous study, we collected the questionnaire on the frequency of defecation (per day) and shape of stools scored as seven grades in a Bristol stool chart [20].

### 2.4. DNA Extraction and 16S rRNA Gene Amplicon Sequencing

The fecal sample mixture was mechanically disrupted using the bead-beating method. DNA was extracted using Gene Prep Star PI-80X (Kurabo Industries, Osaka, Japan). After DNA extraction, the V3–V4 region of the 16S rRNA gene was amplified using the following primers: forward, 5′-TCG GCA GCG TCA GAT GTG TAT AAG CGA CAG CCT ACG GGN GGC WGC AG-3′; reverse, 5′-GTC TCG TGG GCT CGG AGA TGT GTA TAA GAG ACA GGA CTA CHV GGG TAT CTA ATC C-3′ [29]. Amplicons were sequenced by the paired-end method using MiSeq (Illumina, San Diego, CA, USA). The overall procedure, from fecal sampling to 16S rRNA sequencing, was performed according to a previously described protocol [30].

### 2.5. Bioinformatics Analysis

The paired-end output from MiSeq was trimmed and merged before operational taxonomic units (OTUs) were selected. The QIIME pipeline (v. 1.9.1) was used to perform OTU classification and diversity analyses [31]. All steps, from trimming to diversity analysis, were automatically performed according to previously described methods [32]. The USEARCH algorithm was used to cluster OTUs against the SILVA 128 reference database [33] at 97% similarity [34]. Taxonomic classification was performed using the SILVA 128 reference database from the phylum to the genus level.

### 2.6. Blood Biochemicals

We adopted data on blood biochemistry to measure conventional risk factors for non-communicable diseases, such as hemoglobin A1c, triglyceride, and total and high-density lipoprotein cholesterol, from the source cohort [20,22]. In brief, blood samples were taken from the participants, and then the samples were subjected to measurements of blood fractionation and serum biomarkers using standard laboratory tests [35]. The list of blood biochemical is summarized as 17 components of “risks” at the end of Appendix A.

### 2.7. Dichotomous Grouping for Ramen Intake

In terms of ramen intake, we defined the ramen group and no-intake group based on a threshold around the median intake of ramen in the study population. According to the distribution of ramen intake, many participants who did not eat ramen at all were identified (Appendix A). A group-to-group comparison of ramen intake from 1 g/1000 kcal/day to 30 g/1000 kcal/day showed a significant difference in the Shannon index between the group that did not eat ramen at all and the group that did (Figure 3A and Appendix A). We defined these two groups as the no_intake and intake groups at the threshold of 1 g/1000 kcal/day of ramen intake, respectively.

Regarding ramen intake grouping, we tried to identify the threshold of the ramen intake that causes the difference in the gut microbial diversity by comparing ramen intake using the threshold from 1 g/1000 kcal/day to 30 g/1000 kcal/day. The results revealed that a significant difference could be found between the group that consumed ramen and the group that did not eat at all (Figure 3A and Appendix A). In fact, according to the distribution of ramen intake, there are many participants who did not eat ramen (Appendix A). Based on this observation, we defined the two groups into ramen intake and ramen no intake (those who did not consume ramen).

### 2.8. Statistical Analysis

For the fecal microbiome analyses, we performed the data analysis according to our previous study [20,22]. In brief, the output of the QIIME pipeline in the Biom table format was imported and analyzed using R (version 3.5.1). The alpha diversity index was calculated using the estimate_richness function in the “phyloseq” R package. For correlation analysis, the dominant bacteria from the phylum to the genus level were defined as those with an average bacterial composition of at least 1%.

We performed a Spearman correlation analysis between the fecal microbiota and the metadata. The metadata includes 58 food intakes (enumerated in Appendix A) and 3 alpha diversity indices (Shannon, Simpson, and Fisher’s alpha). The cor function in the R package “stats” was used in this analysis. Based on the ramen intake, we further performed the Wilcoxon rank sum test (wilcox.test function in “stats” R package) to find differences in the nutrient intakes, the observed fecal microbiome data, and the Shannon index between the ramen group and the no-intake group. A z-score is calculated as a measure of how far a value is above or below the mean of a data set in each variable. Additionally, a multiple linear regression analysis by forward selection was used to identify food intake associated with the Shannon index using the two bowel habits, fifty-eight foods, and one risk (BMI) from all applicable variables shown in Appendix A (i.e., lm function and step function in the R package “stats”). A stepwise multiple linear regression model was analyzed using the Shannon index as a dependent variable and food intake, age, BMI, defecation frequency, and gut stool shape as independent variables. Defecation frequency and gut stool shape were added as independent variables because they were confirmed to be related to intestinal bacteria in our previous study [20]. The dependent variable was standardized. Another regression model was developed for ramen, and the Shannon index was adjusted with age, BMI, defecation frequency, and stool shape.

To explore possible associations between dietary ramen exposure and a diverse variable from the dataset in the NEXIS cohort study, a multiple logistic regression analysis by forward selection was used to identify a series of variables, including 3 anthropometrics, 2 bowel habits, 50 nutrients, 23 risks and 77 gut microbiota (as described in Appendix A) associated with the ramen intake group (i.e., glm function and step function in the R package “stats”).

Statistical analyses were performed using R software (version 3.5.1). All statistical tests were considered significant differences with a significance level of *p* < 0.05. Heatmaps were created using the “superheat” R package, and the other graphs were created using the R package “ggplot2.”

## 3. Results

### 3.1. Gut Microbiota Diversity Associated with Ramen Intake

To clarify the effect of food intake on gut microbial diversity, the correlation between food intake estimated from dietary intake history and the alpha diversity index calculated from gut microbiota community data obtained by 16SrRNA amplicon sequencing was analyzed in a cross-sectional study of 224 healthy Japanese female subjects. The alpha diversity indicator is defined as having more diversity if the value is higher. Ramen intake showed a statistically significant negative correlation with Shannon and Simpson diversity indices (Figure 1A). Both correlation coefficients were −0.15, with similar *p*-values (Shannon index, *p* = 0.027; Simpson diversity index, *p* = 0.030) (Figure 1B,C). No food group other than ramen showed any statistically significant correlation with the alpha diversity index. As a result of checking the eating habits related to ramen by performing a correlation analysis between foods, noodles soup, and noodles such as udon, pasta, and soba were most related to ramen, whereas rice crackers, alcohol types such as beer, wine, and shochu, and beverages such as coke and coffee were also modestly interrelated with ramen in a large cluster. Conversely, the opposite eating habits were identified for vegetables other than pickles, tofu, fried tofu, natto, fatty fish, and small fish (Appendix A).

We performed an analysis considering multicollinearity to further clarify the relationship between alpha diversity and ramen intakes obtained from the correlation analysis. As predictors of the Shannon index, ramen intake was extracted as a negative factor, and Shochu, pasta, wine intake, and age were extracted as positive factors using the stepwise method (Figure 2). In this model, ramen intake was significantly inversely correlated, and Shochu intake, stool frequency, and age were significantly positively correlated with the Shannon index (β = −0.018 for ramen intake, β = 0.005 for Shochu intake, β = 0.08 for gut habit stool frequency, and β = 0.008 for age). Among these factors, ramen intake was found to be the most influential on the Shannon index. The relationship between ramen intake and Shannon index was also confirmed to be significant in the multiple linear regression model by setting age, BMI, defecation frequency, and stool shape as confounding factors (*p* = 0.032) (Appendix A).

### 3.2. Differences in Gut Bacteria and Nutrient Intake Due to Ramen Intake

We confirmed that the ramen intake is a variable that independently affects the diversity of gut bacteria. Therefore, we focused on the ramen intake variable and then proceeded with the analysis. The characteristics of phenotype and health indicators using blood data between the Raman intake and no_intake groups are expressed in Appendix A. There was a difference in age and height, but there was no difference other than that.

We first conducted a comparative analysis between the two groups to determine which gut bacteria among the various gut bacteria are related to ramen intake. We compared the differences in gut bacteria between the two groups according to the ramen intake by genus (Figure 3B, Appendix A). It was analyzed for gut bacteria with an average value of 0.1% or more of relative abundance. In the ramen intake group, bacterial genera such as *Phascolarctobacterium*, *Dorea*, *Providencia*, *Eubacterium eligens* group, and *Anaerostipes* showed significantly lower z-scores than those in the no_intake group.

In Japanese dietary culture, ramen is generally regarded as a proper meal rather than a concept of snacks or late-night snacks. In addition, when ramen is generally consumed as a meal, the intake of foods other than ramen is limited. Therefore, we assumed that various nutrient intakes were lacking. The two groups’ comparative nutrient intake analysis revealed significant differences in various nutrients (Figure 3C). For example, the intake group had a significantly lower intake of vitamins, minerals, and dietary fiber than the no_intake group. Specifically, significantly lower intake was observed for vitamin C, vitamin B1, vitamin B6, and pantothenic acid among water-soluble vitamins; for α-tocopherol, vitamin K, and the provitamin A carotenoid β-carotene among fat-soluble vitamins; and for potassium, magnesium, zinc, and copper among minerals. For dietary fiber, soluble and total fiber intake were low (Figure 3C, Appendix A). Daidzein, genistein, and isoflavones were also significantly lower. On the other hand, there was no difference in the intake of major nutrients such as total energy intake, protein intake, and fat intake, carbohydrate intake between the two groups. In addition, alcohol intake did not show any difference between the two groups.

### 3.3. Multivariate Analysis for Ramen Intake Patterns and Nutrient Intake, Gut Microbiota, and Blood Biochemical Indices

Finally, we performed a multivariate logistic regression analysis using all nutrition intake data, age, BMI, defecation frequency, gut stool shape, gut bacteriological community, and blood biochemical indicators to assess how ramen intake affects health and to identify independent nutrients and gut bacteria. Odds ratios and 95% confidence intervals were calculated (Table 2). The results showed that some of the factors were different compared to the nutrients and gut bacteria extracted in the two-group comparison, as shown in Figure 3. Among the variables incorporated in the regression model, ash content intake (adjusted odds ratio = 1.06 × 10^10^, *p* = 0.006), pantothenic acid intake (adjusted odds ratio = 8.37 × 10^4^, *p* < 0.001), and n-3 polyunsaturated fatty acid intake (adjusted odds ratio = 3.70 × 10^4^, *p* < 0.001) were factors with less than significance levels and high odds ratios. Age and defecation frequency was related to the presence or absence of ramen consumption, respectively (adjusted odds ratio = 1.17, *p* = 0.004; adjusted odds ratio = 0.24, *p* = 0.002). In contrast, gut stool shape was not confirmed to have an independent relationship with the presence or absence of ramen. Among gut bacteria, the relationship between various bacteria belonging to the Lachnospiraceae and Ruminococcaceae families, including *Faecalibacterium*, *Dorea*, and *Roseburia*, was prominent. In addition, *Bifidobacterium*, *Barnesiella*, and *Prevotella.2*, *Peptoclostridium*, *Holdemanella*, and *Proteus* were independently associated with the presence or absence of ramen intake. In the nutrient category, it was confirmed that the intake of protein and dietary fiber, along with total energy intake, was related to the presence or absence of ramen intake, and various trace nutrients were confirmed. Blood biochemical indicators between the ramen intake group and the no_intake group did not show any significant difference as a result of comparative analysis through the Wilcoxon rank sum test (Appendix A), however as a result of multivariate logistic regression analysis, total cholesterol (adjusted odds ratio = 0.96, *p* = 0.002), diastolic blood pressure (adjusted odds ratio = 1.08, *p* = 0.049), γ-glutamyl transferase (adjusted odds ratio = 1.07, *p* < 0.001), and mean corpuscular volume (adjusted odds ratio = 1.84, *p* < 0.001) were confirmed to be related to the presence or absence of ramen intake.

## 4. Discussion

Interpretation of the relationship between food intake and gut microbiome in a healthy Japanese female cohort revealed that ramen intake correlated with gut bacterial alpha diversity, and ramen intake was associated with a distinctive dietary pattern. In addition, ramen intake habits can affect a series of dietary nutrients, specific gut bacteria, and blood biochemical indicators.

This study extends the inverse link between ramen intake and alpha diversity from postmenopausal women [24] to healthy female adults in the Japanese population. Although not detailed in this report, a comparative analysis between hierarchical classification groups using meal data showed no difference in the diversity of intestinal bacteria. However, in the hierarchical clustering correlation matrix, food intake was thought to be negatively correlated with meals with relatively diverse nutrient intakes, such as noodles, desserts, and beverages, which was confirmed as a characteristic of the dietary pattern related to ramen intake (Appendix A). Based on the classification of dietary patterns in a population-based study in Japan, beverages, bread, and snacks with noodles and sugar, including ramen, were classified in the same cluster [36]. As this study also confirmed the relationship between similar food intake, it is suggested that the typical characteristics of Japanese women’s eating habits were reflected in the results of the study.

In the comparative analyses with respect to ramen intake, the dietary pattern was characterized by low vitamin and fiber intake in the ramen intake group. According to well-established knowledge, dietary fiber promotes a healthy gut microbiome, while some vitamins and minerals also influence microbiome formation. Pantothenic acid, potassium, vitamin K, and magnesium had the highest z-score > 0.25 with <0.05 when compared to ramen intake. Lack of pantothenic acid adversely affects the immune system and induces a pro-inflammatory state [37], whereas high dietary potassium can be characterized as a vegetable- and fruit-rich diet with an abundance of dietary fiber [38]. Dietary vitamin K is derived from the habitual intake of seafood and fermented soybeans [39]. Consequently, frequent ramen intake appeared to be associated with a pro-inflammatory dietary pattern, including low pantothenic acid, potassium, vitamin K, and dietary fiber, possibly due to the limited intake of fruits, vegetables, and seafood. In this study, the dietary pattern with reduced ramen intake influenced the relative abundance of SCFA-producing bacteria, such as Dorea.

After adjusting for age, BMI, defecation frequency, and gut stool shape as independent variables, the multivariate linear regression model predicted that Shochu liquor intake, along with ramen as a negative factor, had a positive association with the Shannon index. According to the literature, the relationship between alcohol intake and the gut microbiome is not straightforward. Rhesus monkeys given alcohol showed decreased alpha diversity and increased Firmicutes in the gut microbiome, as well as an impaired metabolome during glycolysis [40]. Regarding the dose-dependency of alcohol on alpha diversity, light consumers have the highest Shannon and Simpson indices among never, light, and heavy consumers [41]. The median intake of Shochu liquor was 5.81 g/1000 kcal/day in this study, resulting in a moderate intake of Shochu liquor as a positive factor in the Shannon index.

In this study, ramen intake was significantly associated with the alpha diversity of the gut bacterial community (Figure 3A), as previously reported [24]. In addition to diversity, ramen intake showed an effect on the relative abundance of *Dorea* in simple comparative and multivariate analysis results, while the direction of association was inconsistent (Figure 3C, Table 2). *Dorea* is a SCFA-producing gut microbiota. In recent years, it has been reported that SCFAs regulate GPR41 and GPR43 and have many health functions [42,43]. *Dorea longicatena* is a producer of indole-3-acetate, and indole-3-acetate involves anti-inflammatory and anti-oxidative activity in in vitro experiments [44,45]. From the viewpoint of diet and *Dorea*, the genus *Dorea* was overexpressed in diets with higher polyunsaturated fatty acids/saturated fatty acids ratio [46]. Taken together, the literature partly supported the same direction of association in n-3 polyunsaturated fatty acids and *Dorea* in the ramen group in a multivariate logistic model, although the diet in the ramen group appeared to be inconsistent with the anti-oxidative and anti-inflammatory diet in this study.

One of the features of this study is that blood markers were included in the association between diet and gut microbiota. In fact, γ-GTP, a liver function marker, was significantly worse in the ramen intake group than in the no_intake group (Table 2). Dietary patterns may be involved in serum γ-GTP levels in a large population-based cross-sectional study in Japanese men and women [47], implying that clinical markers of liver diseases can be associated with dietary patterns in healthy Japanese females. In this study, ramen intake was associated with SCFA-producing gut microbiota and γ-GTP in the multivariate analysis adjusted by nutrient intake, gut microbiota, and blood biochemical indices. Taken together, frequent consumption of ramen shapes a dietary feature along with gut microbiome and diet-susceptible blood biomarkers. Considering the dietary patterns observed in this study, advocacy comprising less sugar and more minerals, vitamins, fibers, and n-3 polyunsaturated fatty acids may be a potential and practical application for dietary management in frequent ramen consumers.

A limitation of this study is that it was a cross-sectional study, and the relationship between food intake and gut microbiota or other blood indices is unclear. As this study was conducted in a single country, region, and race, its reproducibility should be confirmed in different countries and/or races. The BDHQ estimates the nutritional components of a small number of nutrients. Furthermore, a questionnaire regarding nutritional compounds during the preceding months was flawed due to uncertainties. We did not distinguish between sugar and artificial sweeteners for the intake of sugar-added beverages using a food frequency questionnaire. Emerging evidence suggests that the intake of artificial sweeteners is associated with obesity [48]. Moreover, sweetened beverages influence the gut microbiome regardless of the type of sweetener, such as sugar or artificial sweeteners [49]. Although the intake of sugar-added beverages was not significantly correlated with the Shannon index in this study, the sugar-added beverage could be a potential confounder in the associations between the ramen diet and gut microbiome in this study. For the limitations in statistical models, we adopted the stepwise modeling for exploratory analysis, although this model has potential biases and shortcomings due to bias in parameter estimation, inconsistencies in model selection algorithms, an inherent problem of multiple testing, and an inappropriate focus or reliance on the single best model [50]. Further study is warranted to demonstrate the variables associated with the ramen group in this study.

## 5. Conclusions

In this study, the relationship between specific eating habits and gut bacterial diversity in a healthy Japanese female cohort was analyzed, and the relationship with health indicators was analyzed using dietary nutrient intake, gut microbiome, and blood biochemistry, including blood pressure. Ramen intake inversely correlated with intestinal bacterial alpha diversity and showed distinguished dietary patterns compared to the participant who consumed less ramen. It was confirmed that foods that affected general Japanese meals, primarily ramen, were related to alpha diversity. It has been confirmed that these eating habits have a negative effect on the consumption of various nutrients, which can also affect intestinal bacteria and health. Although further demonstration in an intervention trial still is needed because of the nature of the cross-sectional design of this study, people with frequent ramen eating habits need to take measures to consume various nutrients to maintain and improve their health, as well as dietary management that can be applied to the dietary feature in ramen consumption. The results of this study provide insights into eating habits to improve the intestinal environment and health.

## Figures and Tables

**Figure 1 microorganisms-11-01892-f001:**
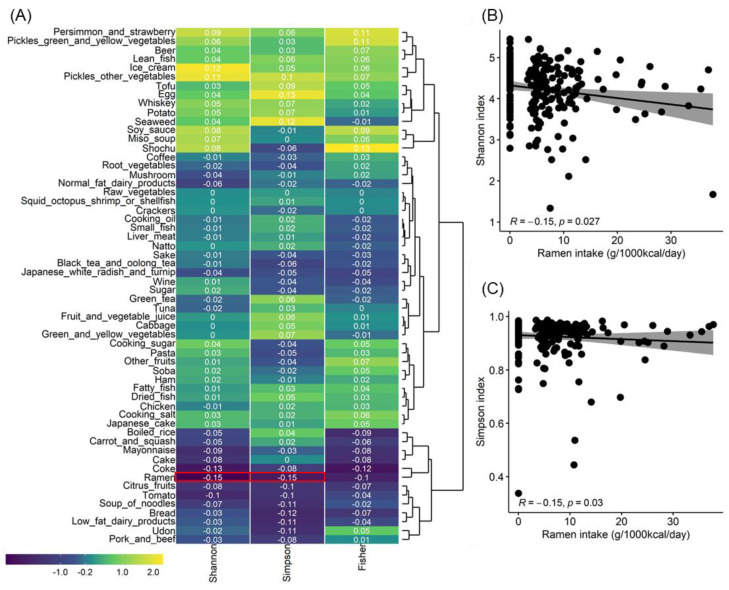
Correlation analysis results between gut bacterial alpha diversity and food intake. (**A**) This heat map shows the results of the correlation analysis between gut bacterial alpha diversity and food intake. The colors are coded according to Spearman’s correlation coefficient, with the red box representing a significant correlation (*p* value < 0.05). (**B**) Scatter plots based on the Shannon index and ramen intake. (**C**) Scatter plots based on the Simpson index and ramen intake.

**Figure 2 microorganisms-11-01892-f002:**
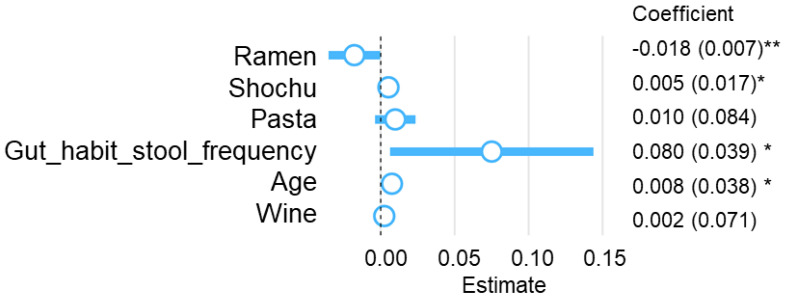
Stepwise multiple linear regression analysis for Shannon index. Coefficients are unstandardized OLS partial regression slopes with standard errors in parentheses. Blue circles represent the estimated coefficients. ** *p* value < 0.01, * *p* value < 0.05.

**Figure 3 microorganisms-11-01892-f003:**
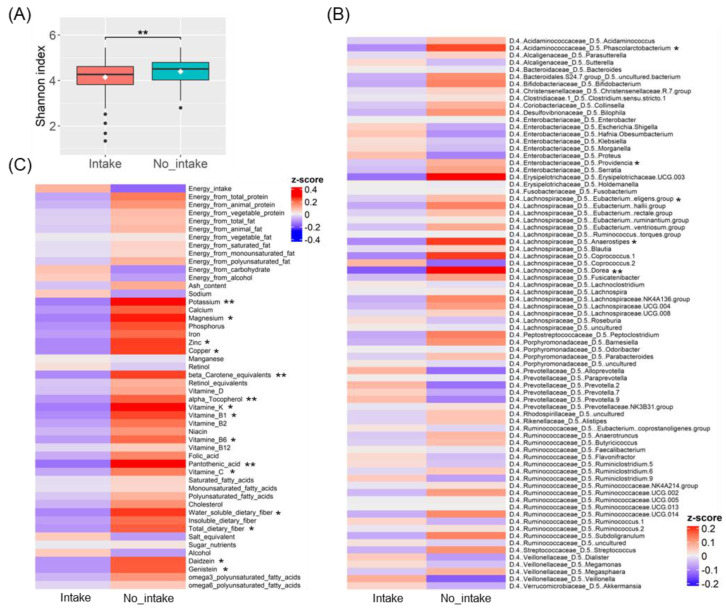
Comparison between two groups separated according to ramen intake. (**A**) Comparison of Shannon index between intake and no_intake groups using the Wilcoxon rank sum test. (**B**) A z-score heat map comparing bacterial genus relative abundance between intake and no_intake groups using the Wilcoxon rank sum test. (**C**) A z-score heat map comparing nutrient intake between intake and no_intake groups using the Wilcoxon rank sum test. ** *p* value < 0.01, * *p* value < 0.05.

**Table 1 microorganisms-11-01892-t001:** Background characteristics in the study population.

Variables	Units	Mean ± SD	Variables	Units	Mean ± SD
Demographic and anthropometrics			Hematocrit	%	40.2 ± 2.8
Age	y	59.5 ± 11.9	Platelet	10^4^ count/μL	2.39 ± 4.8
Body height	cm	156.5 ± 5.8	Mean corpuscular volume (MCV)	fl	93.1 ± 4.7
Body weight	kg	54.6 ± 7.4	Mean corpuscular hemoglobin (MCH)	pg	30.3 ± 1.9
BMI	kg/m^2^	22.3 ± 2.9	Mean corpuscular hemoglobin concentration (MCHC)	%	32.6 ± 1.0
Waist circumference	cm	80.4 ± 9.3	Aspartate transaminase (AST)	IU/L	22.1 ± 5.8
Hip circumference	cm	92.1 ± 5.4	Alanine transaminase (ALT)	IU/L	16.7 ± 7.4
Calf circumference	cm	34.3 ± 2.3	γ-Glutamyltransferase (γ-GTP)	IU/L	23.9 ± 19.2
Blood pressure and Blood biochemistry			Fasting glucose	mg/dL	88.3 ± 15.3
Systolic blood pressure	mmHg	120.0 ± 17.1	Hemoglobin A1c (HbA1c)	%	5.57 ± 0.51
Diastolic blood pressure	mmHg	71.2 ± 10.5	Triglyceride	mg/dL	86.6 ± 49.6
White blood cell (WBC)	count/μL	4681 ± 1135	Total cholesterol (Cho)	mg/dL	219.4 ± 35.9
Red blood cell (RBC)	10^4^ count/μL	433 ± 32	High-density lipoprotein cholesterol (HDL-cho)	mg/dL	72.6 ± 18.1
Hemoglobin	g/dL	13.1 ± 1.1	Fasting insulin	μU/dL	4.09 ± 4.01

**Table 2 microorganisms-11-01892-t002:** Multivariate analysis for ramen intake patterns and nutrient intake, gut microbiota, and blood biochemical indices.

	Crude OR (95%CI)	Adj. OR (95%CI)	P (Wald’s Test)
Age	1.03 (1.01, 1.06)	1.17 (1.05, 1.3)	0.004
Gut_habit_stool_frequency	0.93 (0.73, 1.18)	0.24 (0.1, 0.6)	0.002
D.1..Actinobacteria_D.2..Actinobacteria_D.3..Bifidobacteriales_D.4..Bifidobacteriaceae_D.5..Bifidobacterium	1.0003 (0.9999, 1.0008)	1.0013 (1, 1.0026)	0.047
D.1..Bacteroidetes_D.2..Bacteroidia_D.3..Bacteroidales_D.4..Porphyromonadaceae_D.5..Barnesiella	1 (1, 1)	1.01 (1, 1.02)	0.032
D.1..Bacteroidetes_D.2..Bacteroidia_D.3..Bacteroidales_D.4..Prevotellaceae_D.5..Alloprevotella	0.95 (0.86, 1.06)	0.72 (0.33, 1.57)	0.405
D.1..Bacteroidetes_D.2..Bacteroidia_D.3..Bacteroidales_D.4..Prevotellaceae_D.5..Prevotella.2	0.9983 (0.9955, 1.0011)	0.9937 (0.9891, 0.9984)	0.009
D.1..Firmicutes_D.2..Clostridia_D.3..Clostridiales_D.4..Lachnospiraceae_D.5...Eubacterium..ruminantium.group	1 (1, 1)	1.01 (1, 1.02)	0.016
D.1..Firmicutes_D.2..Clostridia_D.3..Clostridiales_D.4..Lachnospiraceae_D.5...Eubacterium..ventriosum.group	1 (1, 1.01)	1.02 (1, 1.04)	0.038
D.1..Firmicutes_D.2..Clostridia_D.3..Clostridiales_D.4..Lachnospiraceae_D.5..Dorea	1.02 (1.01, 1.03)	1.12 (1.07, 1.17)	<0.001
D.1..Firmicutes_D.2..Clostridia_D.3..Clostridiales_D.4..Lachnospiraceae_D.5..Lachnoclostridium	1 (1, 1)	0.98 (0.98, 0.99)	<0.001
D.1..Firmicutes_D.2..Clostridia_D.3..Clostridiales_D.4..Lachnospiraceae_D.5..Lachnospiraceae.NK4A136.group	1.0021 (0.9989, 1.0054)	1.0056 (0.9976, 1.0137)	0.171
D.1..Firmicutes_D.2..Clostridia_D.3..Clostridiales_D.4..Lachnospiraceae_D.5..Roseburia	0.9998 (0.999, 1.0006)	0.9974 (0.9952, 0.9995)	0.017
D.1..Firmicutes_D.2..Clostridia_D.3..Clostridiales_D.4..Peptostreptococcaceae_D.5..Peptoclostridium	1.01 (0.99, 1.02)	1.13 (1.07, 1.2)	<0.001
D.1..Firmicutes_D.2..Clostridia_D.3..Clostridiales_D.4..Ruminococcaceae_D.5..Faecalibacterium	1 (0.9997, 1.0003)	0.9985 (0.9976, 0.9993)	<0.001
D.1..Firmicutes_D.2..Clostridia_D.3..Clostridiales_D.4..Ruminococcaceae_D.5..Ruminiclostridium.9	0.99 (0.97, 1.01)	0.93 (0.89, 0.98)	0.009
D.1..Firmicutes_D.2..Clostridia_D.3..Clostridiales_D.4..Ruminococcaceae_D.5..Ruminococcaceae.NK4A214.group	1 (0.99, 1.01)	0.96 (0.94, 0.98)	<0.001
D.1..Firmicutes_D.2..Clostridia_D.3..Clostridiales_D.4..Ruminococcaceae_D.5..Ruminococcaceae.UCG.014	1 (1, 1)	1.02 (1.01, 1.03)	<0.001
D.1..Firmicutes_D.2..Clostridia_D.3..Clostridiales_D.4..Ruminococcaceae_D.5..Ruminococcus.1	0.9993 (0.9968, 1.0017)	0.9907 (0.9857, 0.9958)	<0.001
D.1..Firmicutes_D.2..Clostridia_D.3..Clostridiales_D.4..Ruminococcaceae_D.5..uncultured	1 (1, 1)	1.01 (1, 1.02)	0.009
D.1..Firmicutes_D.2..Erysipelotrichia_D.3..Erysipelotrichales_D.4..Erysipelotrichaceae_D.5..Erysipelotrichaceae.UCG.003	1.01 (1, 1.02)	1.02 (0.99, 1.05)	0.141
D.1..Firmicutes_D.2..Erysipelotrichia_D.3..Erysipelotrichales_D.4..Erysipelotrichaceae_D.5..Holdemanella	1 (1, 1)	0.98 (0.98, 0.99)	<0.001
D.1..Firmicutes_D.2..Negativicutes_D.3..Selenomonadales_D.4..Veillonellaceae_D.5..Megamonas	0.9998 (0.9992, 1.0005)	0.9991 (0.9979, 1.0003)	0.125
D.1..Proteobacteria_D.2..Alphaproteobacteria_D.3..Rhodospirillales_D.4..Rhodospirillaceae_D.5..uncultured	1.0008 (0.998, 1.0035)	1.0084 (1.002, 1.0149)	0.01
D.1..Proteobacteria_D.2..Gammaproteobacteria_D.3..Enterobacteriales_D.4..Enterobacteriaceae_D.5..Proteus	0.98 (0.91, 1.05)	0.57 (0.3, 1.1)	0.096
Energy_intake	0.9993 (0.9986, 1)	0.9947 (0.9922, 0.9973)	<0.001
Energy_from_total_protein	1.1 (1, 1.22)	0.22 (0.05, 0.94)	0.041
Insoluble_dietary_fiber	1.21 (1.01, 1.44)	0.11 (0.03, 0.43)	0.002
Ash_content	1.09 (0.94, 1.26)	1.06 × 10^10^ (785.16, 1.43 × 10^10^)	0.006
Beta_Carotene_equivalents	1.0002 (1.0001, 1.0004)	1.002 (1.0009, 1.0031)	<0.001
Calcium	1 (1, 1.01)	0.92 (0.89, 0.96)	<0.001
Pantothenic_acid	1.95 (1.29, 2.97)	8.37 ×10^4^ (352.47, 1.99 ×10^7^)	<0.001
Phosphorus	1 (1, 1.01)	1.05 (1, 1.1)	0.054
Potassium	1 (1, 1)	0.96 (0.93, 1)	0.03
Sodium	1 (1, 1)	0.94 (0.9, 0.98)	0.002
Sugar	1.01 (0.94, 1.08)	1.63 (1.27, 2.1)	<0.001
Vitamine_B2	4.49 (1.01, 19.95)	0 (0, 0)	<0.001
Vitamine_B6	5.72 (1.11, 29.35)	0 (0, 0.01)	0.017
omega3_polyunsaturated_fatty_acids	1.64 (0.85, 3.17)	3.70 × 10^4^ (340.09, 4.03 × 10^6^)	<0.001
Total_cholesterol	1 (0.99, 1.01)	0.96 (0.94, 0.99)	0.002
Diastolic_blood_pressure	1.01 (0.99, 1.04)	1.08 (1, 1.16)	0.049
gamma_Glutamyltransferase	1.01 (0.99, 1.02)	1.07 (1.03, 1.12)	<0.001
Mean_corpuscular_volume	1.07 (1, 1.14)	1.84 (1.39, 2.42)	<0.001

## Data Availability

The human gut sequencing data from this study have been deposited in the DDBJ Sequence Read Archive under the accession number DRP007218.

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
