# Peer review of "Ramen Consumption and Gut Microbiota Diversity in Japanese Women: Cross-Sectional Data from the NEXIS Cohort Study"

_microorganisms, 2023, doi:10.3390/microorganisms11081892_

Round 1

Reviewer 1 Report (New Reviewer)

Author Response

  1. Line 59- please edit since the word Ramen is used twice

As you pointed out, we removed the repetition of Ramen for the Keywords in Line 62 in the revised manuscript.

  1. Table 1 - I recommend that the authors consider editing the table to improve its comprehensibility and clarity. One possible approach could be to condense the data into four columns or a similar format. By doing so, the information presented in the table would be more concise and easier to understand. This adjustment would enhance the overall readability and effectiveness of the table.

We organized the table layout as shown in the four columns and two variables in a row. Please find Table 1 in the revised manuscript.

  1. Line 152-153: Please revise the sentence since its intended meaning is unclear and has been lost.

We clarified the methods to review and correct the blank and incorrect answers in a questionnaire. Please review the revision in Lines 180-181 in the revised manuscript as follows:

A face-to-face interview was optionally conducted with the participants to review and correct the missing and inappropriate answers.

  1. Line 191-192 – please, the sentence needs revision as it is unclear whether the participants did not consume Ramen or any other food.

We distinguished the participants who did not eat Ramen in a dichotomous manner. I corrected the sentence in Lines 223-225 in the revised manuscript as follows:

According to the distribution of Ramen intake, many participants who did not eat Ramen at all were identified (Figure S2A, Figure S2B).

  1. Lines 205-208 – please revise this text in order to make it clearer and more understandable.

We reviewed the original description and revised it for clarity in the correlation analyses between the alpha diversity index and food intake in Lines 245-248 as follows:

We performed a Spearman correlation analysis between the fecal microbiota and the metadata. The metadata includes 58 food intakes (enumerated in Table S1) and three alpha diversity indices (Shannon, Simpson, and Fisher’s alpha). The cor function in the R package “stats” was used in this analysis.

  1. Line 430-431 – I kindly request the author to provide further clarification regarding their intended meaning in this statement, and to revise it for better understandability.

Line 476-478 was the requested sentence for further clarification. We revised the description to improve the statement for better understandability in Lines 476-478 in the revised manuscript as follows:

Ramen intake inversely correlated with intestinal bacterial alpha diversity and showed distinguished dietary patterns compared to the participant who consumed less Ramen.

Reviewer 2 Report (New Reviewer)

The paper provides a valuable cross-sectional analysis from the NEXIS cohort study, examining the association between ramen intake and gut microbiota diversity in a population of Japanese women. However, there are several areas that could be enhanced in the study. Here are a few examples of potential improvements:

1:The title is a bit lengthy and wordy. It could be revised to enhance clarity and conciseness: Title Suggestion: "Ramen Consumption and Gut Microbiota Diversity in Japanese Women: Insights from the NEXIS Cohort Study"

2: In lines 48, 49 and line 55, 56 Ramen intake was inversely associated with alpha diversity” what authors mean by this association is it increased or decreased alpha diversity.

3: The abstract could provide more context regarding the rationale for studying the association between ramen intake and gut microbiota diversity, as well as its relevance to non-communicable diseases or the existing body of research.

4. The abstract mentions the use of a stepwise logistic regression model for adjusting related factors but does not specify what these factors are. Including a brief mention of the variables adjusted for in the analysis would be helpful.

5. Error: "whreas" - This seems to be a typographical error. It should be "whereas."line 56

6. The abstract states that individuals with frequent ramen eating habits should take measures to consume various nutrients for health maintenance and improvement. It would be beneficial to provide more specific recommendations or suggestions for achieving a balanced diet in this context.

7. The abstract could briefly discuss the implications of the findings in terms of potential interventions or recommendations for promoting gut health and reducing the risk of non-communicable diseases.

8. While the abstract highlights the inverse association between ramen intake and gut microbiota diversity, it would be helpful to mention the effect size or magnitude of this association, if available increased or decreased.

9. please Provide a concise summary of the limitations of the study, such as the cross-sectional design, reliance on self-reported ramen intake, or any other potential sources of bias or confounding.

10-Remember to carefully proofread the manuscript to eliminate any typographical, grammatical errors and ensure clarity in conveying the study's key message.

11- Please check the references there are some mistakes in the manuscript.

12- The introduction could provide more specific information about the role of gut microbiota in human health, such as the importance of maintaining a diverse and balanced gut microbiome for optimal physiological functioning and disease prevention.

13- While the introduction mentions the low obesity rate and distinctive diet in Japan, it would be beneficial to provide a brief explanation or examples of the components or characteristics of the Japanese diet that contribute to its health benefits.

14- The introduction states that recent findings have demonstrated the role of the intestinal environment in human health, but it does not elaborate on these findings or provide specific references to support this claim, including a summary of key studies or findings.

15- The introduction highlights the association between diet and gut bacterial community structure but does not clearly specify the research gap in this area.

16- Is not only bacterial communities that affect host physiology there are other types such as so it should be noted in the manuscript and acknowledge the limitation and correct this in the manuscript as well.

17- The introduction briefly mentions the analysis of the relationship between intestinal bacterial diversity, dietary intake patterns, and urinary equol concentrations in postmenopausal Japanese women. However, it does not provide a clear reason for including blood data as a health indicator.

18- It is important to state why ramen intake was chosen and its relevance to the research question would enhance the introduction's clarity.

19- A total of the 954 healthy Japanese adults was included in the source cohort" - The phrase "the" before "954" seems to be a typo and should be removed.

20- The method section could provide more information about the inclusion and exclusion criteria for participant selection, including specific details about the criteria for healthy adult women.

21- The method section mentions that a brief self-administered diet history questionnaire (BDHQ) was used for assessing dietary habits. It would be beneficial to briefly describe the content and format of the BDHQ to provide readers with an understanding of the questionnaire's scope and purpose.

22- It would be helpful to mention the specific version of QIIME and any customized settings employed for the analysis and what R version.

23- The method section states that blood biochemistry measurements were taken, but it does not provide a comprehensive list of the specific biomarkers or tests performed.

24- The method does not provide the specific values used for defining the ramen and no-intake groups. Adding this information would make the methodology more explicit.

25- When discussing the differences in gut bacteria between the Ramen intake and no-intake groups, the results section mentions bacterial genera such as Phascolarctobacterium, Dorea, Providencia, Eubacterium eligens group, and Anaerostipes. However, it would be helpful to provide the direction of the differences (i.e., higher or lower abundance) for each bacterial genus to facilitate interpretation.

 26- why the observed associations with Dorea and SCFA-producing bacteria are noteworthy and discuss their potential implications for health in the manuscript

27- Explain the associations among Ramen intake, gut microbiota, and blood markers in the discussion

28- Explain the limitations of the study, such as the cross-sectional design and the need for further research, including longitudinal studies and investigations in diverse populations

Remember to carefully proofread the manuscript to eliminate any typographical, or grammatical errors and ensure clarity in conveying the study's key message.

Author Response

  1. The title is a bit lengthy and wordy. It could be revised to enhance clarity and conciseness: Title Suggestion: "Ramen Consumption and Gut Microbiota Diversity in Japanese Women: Insights from the NEXIS Cohort Study"

The title was shortened and summarized for clarity and conciseness by the suggestion. Please review the revised title as follows:

Ramen consumption and gut microbiota diversity in Japanese women: cross-sectional data from the NEXIS Cohort Study

  1. In lines 48, 49 and line 55, 56 Ramen intake was inversely associated with alpha diversity” what authors mean by this association is it increased or decreased alpha diversity.

We revised the sentence to understand the meaning that Ramen intake decreased alpha diversity in the description in Lines 46-49 and Lines 54-57 in the revised manuscript as follow:

Using a stepwise regression model, Ramen intake was inversely associated with gut microbiome alpha diversity after adjusting for related factors, including diets, Age, BMI, and stool habits (β = -0.018; r = -0.15 for Shannon index).

In conclusion, the increased Ramen intake associated with decreased gut bacterial diversity ac-companying a perturbation of Dorea through the dietary nutrients, gut microbiota, and blood chemistry, while the methodological limitations existed in a cross-sectional study.

  1. The abstract could provide more context regarding the rationale for studying the association between ramen intake and gut microbiota diversity, its relevance to non-communicable diseases or the existing body of research.

We added more concise implications on the key findings from this study in Lines 54-57 in the revised abstract as follows:

“In conclusion, the increased Ramen intake associated with decreased gut bacterial diversity ac-companying a perturbation of Dorea through the dietary nutrients, gut microbiota, and blood chemistry, while the methodological limitations existed in a cross-sectional study.”

Dorea was found to make a variation in Ramen consumption, while the directions of association were still inconsistent in the statistical models. Thus, we indicated Dorea as a gut bacterial genus influenced by Ramen consumption but no straightforward approach of associations in the revised abstract.

  1. The abstract mentions the use of a stepwise logistic regression model for adjusting related factors but does not specify what these factors are. Including a brief mention of the variables adjusted for in the analysis would be helpful.

We added a summary of variables in the abstract in Lines 46-49 and Lines 50-54 in the revised manuscript as follows:

Using a stepwise regression model, Ramen intake was inversely associated with gut microbiome alpha diversity after adjusting for related factors, including diets, Age, BMI, and stool habits (β = -0.018; r = -0.15 for Shannon index).

Sugar intake, Dorea as a short-chain fatty acid (SCFA)-producing gut microbiota, and γ-glutamyl transferase as a liver function marker were directly associated with Ramen intake after adjustment for related factors including diets, gut microbiota, and blood chemistry using a stepwise logistic regression model, whereas Dorea is inconsistently less abundant in the Ramen group.

  1. Error: "whreas" - This seems to be a typographical error. It should be "whereas."line 56

We corrected the word “whereas” in the description in Line 55 in the revised manuscript.

  1. The abstract states that individuals with frequent ramen eating habits should take measures to consume various nutrients for health maintenance and improvement. It would be beneficial to provide more specific recommendations or suggestions for achieving a balanced diet in this context.

Thank you for the insightful comment. Considering the results and discussion, we added the proposed recommendation for achieving a balanced diet in individuals with frequent Ramen eating habits in Lines 57-59, Lines 449-452, and Lines 481-485 in the revised manuscript as follows:

People with frequent Ramen eating habits need to take measures to consume various nutrients to maintain and improve their health, and dietary management can be applied to the dietary feature in Ramen consumption.

Considering the dietary patterns observed in this study, advocacy comprising less sugar and more minerals, vitamins, fibers, and n-3 polyunsaturated fatty acids may be a potential and practical application for dietary management in frequent Ramen consumers.

Although further demonstration in an intervention trial still is needed because of the nature of the cross-sectional design of this study, people with frequent Ramen eating habits need to take measures to consume various nutrients to maintain and improve their health, as well as dietary management can be applied to the dietary feature in Ramen consumption.

  1. The abstract could briefly discuss the implications of the findings in terms of potential interventions or recommendations for promoting gut health and reducing the risk of non-communicable diseases.

In response to comment #6, We added a brief implication from the findings in the context of diet and non-communicable diseases in Lines 57-59 as follows:

People with frequent Ramen eating habits need to take measures to consume various nutrients to maintain and improve their health, and dietary management can be applied to the dietary feature in Ramen consumption.

  1. While the abstract highlights the inverse association between ramen intake and gut microbiota diversity, it would be helpful to mention the effect size or magnitude of this association, if available increased or decreased.

For the significant outcomes, I added the regression and correlation coefficients as the effect size for Ramen intake and gut alpha diversity index in the stepwise logistic regression analysis and crude correlation analysis in Lines 46-49 in the revised manuscript as follows:

Using a stepwise regression model, Ramen intake was inversely associated with gut microbiome alpha diversity after adjusting for related factors, including diets, Age, BMI, and stool habits (β = -0.018; r = -0.15 for Shannon index).

Also, the direction of association was clarified in the revised abstract in Lines 47-50, shown in response#2. Please see also the response#2.

  1. Please Provide a concise summary of the limitations of the study, such as the cross-sectional design, reliance on self-reported ramen intake, or any other potential sources of bias or confounding.

We added the potential limitation arising from the cross-sectional design in Lines 54-57 in the revised manuscript as follows:

In conclusion, the increased Ramen was associated with decreased gut bacterial diversity ac-companying a perturbation of Dorea through the dietary nutrients, gut microbiota, and blood chemistry, while the methodological limitations existed in a cross-sectional study.

  1. Remember to carefully proofread the manuscript to eliminate any typographical, grammatical errors and ensure clarity in conveying the study's key message.

Thank you for your advice. We checked the grammar errors once again and carried out the overall correction.

  1. Please check the references there are some mistakes in the manuscript.

We checked the references and corrected some mistakes.

  1. The introduction could provide more specific information about the role of gut microbiota in human health, such as the importance of maintaining a diverse and balanced gut microbiome for optimal physiological functioning and disease prevention.

According to the comment, we added more specific literature on alpha diversity, physiological status, and disease in Lines 100-105 in the revised introduction as follows:

Lifestyle and clinical factors correlate with Shannon diversity, such as biomarkers for diabetes, inflammation, liver function, and cholesterol [16]. Additionally, n-3 polyunsaturated fatty acids and other markers related to fish intakes, such as docosahexaenoic acid and mercury, are directly correlated with diversity. Diet and physical activity are also correlated with microbiome diversity.

  1. While the introduction mentions the low obesity rate and distinctive diet in Japan, it would be beneficial to provide a brief explanation or examples of the components or characteristics of the Japanese diet that contribute to its health benefits.

Thank you for the suggestion. We added a brief description and typical components of the Japanese diet and their roles in human health in Lines 75-79 and Lines 109-111 in the revised introduction as follows:

The low mortality rate in Japan compared to other countries seems to reflect the low obesity rate, and the low obesity rate is largely due to Japan's distinctive dietary pattern, as characterized by low intake of red meat, high intakes of fish, plant foods, and non-sugar sweetened beverages [5]. A diet comprising fish and n-3 polyunsaturated fatty acids, seaweed, soybean products, and green tea is typically found in washoku, a traditional Japanese diet [18]. Japanese diet partly explains the world’s highest life expectancy from a dietary perspective.

  1. The introduction states that recent findings have demonstrated the role of the intestinal environment in human health, but it does not elaborate on these findings or provide specific references to support this claim, including a summary of key studies or findings.

According to the suggestion, we added a reference to the demonstration as a treatment of intestinal environment improving human health in Lines 105-107 in the revised introduction as follows:

From the viewpoint of clinical intervention on gut microbiota, fecal microbiota transplantation emerged as a promising option for treating Clostridium difficile infection [17].

  1. The introduction highlights the association between diet and gut bacterial community structure but does not clearly specify the research gap in this area.

We have improved the descriptions in the introduction regarding the research gap in the study of the association between diet and gut bacterial community structure.

  1. Is not only bacterial communities that affect host physiology there are other types such as so it should be noted in the manuscript and acknowledge the limitation and correct this in the manuscript as well.

We add the other types of host physiology apart from gut microbiota by the suggestion in Lines 120-123 in the revised introduction as follows:

We thought that dietary habits could influence the highly personalized gut bacterial community and could affect human health complexly with gut bacteria, although other factors could affect the gut bacteria, such as genetic and anthropometric backgrounds, biological aging, and infections.

  1. The introduction briefly mentions the analysis of the relationship between intestinal bacterial diversity, dietary intake patterns, and urinary equol concentrations in postmenopausal Japanese women. However, it does not provide a clear reason for including blood data as a health indicator.

In response to comment #12, comprehensive literature exists on the relationship between alpha diversity and blood biomarkers as a health indicator. Please also refer to response#12 for further information.

  1. It is important to state why ramen intake was chosen and its relevance to the research question would enhance the introduction's clarity.

To enhance the clarity of the introduction and potential relevance to the research question, we added the dietary feature of noodles and its potential impact on the prevention of non-communicable diseases in Lines 131-134 in the revised introduction as follows:

A reduction in dietary sodium intake is a cost-effective public health approach to reduce non-communicable diseases, and noodles, especially instant noodles, are an ultra-processed food that typically contains high sodium from a diet [25].

  1. A total of the 954 healthy Japanese adults was included in the source cohort" - The phrase "the" before "954" seems to be a typo and should be removed.

We removed “the” in Lines 153-157 by your suggestion in the revised methods.

  1. The method section could provide more information about the inclusion and exclusion criteria for participant selection, including specific details about the requirements for healthy adult women.

We revised the sentence to make understanding the inclusion and exclusion criteria for selecting participants easier.

  1. The method section mentions that a brief self-administered diet history questionnaire (BDHQ) was used for assessing dietary habits. It would be beneficial to briefly describe the content and format of the BDHQ to provide readers with an understanding of the questionnaire's scope and purpose.

We added more description regarding the scope and purpose of BDHQ in Lines 166-168 and 175-179 in the revised methods as follows:

Considering the nature of the Japanese diet comprising local seafood, vegetables, and beverages, dietary habits were assessed using a mail-in, brief self-administered diet history questionnaire (BDHQ) [27].

The BDHQ consists of 5 sections: (1) intake frequency of food and nonalcoholic bever-age items generally found in the Japanese diet, (2) daily intake of rice and miso soup as a common combination of staple food and soup in the Japanese diet, (3) frequency of drinking and amount per drink for alcoholic beverages, (4) usual cooking methods in line with Japanese cuisine, and (5) general dietary behavior [28].

  1. It would be helpful to mention the specific version of QIIME and any customized settings employed for the analysis, and what R version.

We added the version information of the tool used for bioinformatics analysis and statistical analysis.

  1. The method section states that blood biochemistry measurements were taken, but it does not provide a comprehensive list of the specific biomarkers or tests performed.

We indicated the comprehensive list of the biomarkers adopted in this study in Lines 217-218 in the revised methods as follows:

The list of blood biochemical is summarized as 17 components of “risks” at the end of Table S1.

  1. The method does not provide the specific values used for defining the ramen and no-intake groups. Adding this information would make the methodology more explicit.

We clarified the threshold for the Ramen intake at 1 g/1000 kcal/day according to the distribution analysis in Lines 227-228 in the revised methods as follows:

We defined these two groups as the no_intake and intake groups at the threshold of 1 g/1000kcal/day of Ramen intake, respectively.

  1. When discussing the differences in gut bacteria between the Ramen intake and no-intake groups, the results section mentions bacterial genera such as Phascolarctobacterium, Dorea, Providencia, Eubacterium eligens group, and Anaerostipes. However, it would be helpful to provide the direction of the differences (i.e., higher or lower abundance) for each bacterial genus to facilitate interpretation.

Thank you for your suggestions. We clarified these bacteria are lower in the Ramen intake group in Lines 324-326 in the revised results as follows:

In the Ramen intake group, bacterial genera such as Phascolarctobacterium, Dorea, Providencia, Eubacterium eligens group, and Anaerostipes showed significantly lower z-scores than those in the no_intake group.

  1. why the observed associations with Dorea and SCFA-producing bacteria are noteworthy and discuss their potential implications for health in the manuscript

Thanks for your comment. In this study, we focused on Dorea because it is the only bacterium with a significant difference in the ramen intake and no_intake groups in simple comparative and multivariate analysis results.

Dorea produces short-chain fatty acids (https://www.ncbi.nlm.nih.gov/pmc/articles/PMC9040766/) and is more abundant in obese individuals than in healthy individuals (https://www.ncbi.nlm.nih.gov/ pmc/articles/PMC9788597/). At first glance, these results seem contradictory, since short-chain fatty acids are known to suppress obesity. Hence, in this case, it is essential to understand the factors that confound Dorea and obesity comprehensively.

  1. Explain the associations among Ramen intake, gut microbiota, and blood markers in the discussion

Thank you for your suggestion. For a summary and possible extension of the association, we added the following sentences in Lines 447-451 as follows:

Taken together, frequent consumption of Ramen shapes a dietary feature along with gut microbiome and diet-susceptible blood biomarkers. Considering the dietary patterns observed in this study, advocacy comprising less sugar and more minerals, vitamins, fibers, and n-3 polyunsaturated fatty acids may be a potential and practical application for dietary management in frequent Ramen consumers.

  1. Explain the limitations of the study, such as the cross-sectional design and the need for further research, including longitudinal studies and investigations in diverse populations

In response to comment #9, I added the potential limitation arising from the cross-sectional design in Lines 54-57 and 480-485 in the revised manuscript as follows:

In conclusion, the increased Ramen was associated with decreased gut bacterial diversity ac-companying a perturbation of Dorea through the dietary nutrients, gut microbiota, and blood chemistry, while the methodological limitations existed in a cross-sectional study.

Although further demonstration in an intervention trial still is needed because of the nature of the cross-sectional design of this study, people with frequent Ramen eating habits need to take measures to consume various nutrients to maintain and improve their health, as well as dietary management can be applied to the dietary feature in Ramen consumption.

Round 2

Reviewer 2 Report (New Reviewer)

NA

NA

This manuscript is a resubmission of an earlier submission. The following is a list of the peer review reports and author responses from that submission.

Round 1

Reviewer 1 Report

In the manuscript entitled "Ramen intake associated with diversity in gut microbiota in a Japanese women cohort", the authors investigated gut microbiome variables from stool samples, several blood-based biomarkers, and subjective reports of dietary intake over the past month. They found that ramen consumption was associated with several measures of microbes and microbiome diversity. The sample of 224 women is impressive and can lead to valuable knowledge.

My major concern is the large variety of variables and tests performed and the lack of clear description and definition of variables and exact models. For example, there is no adequate section or table characterizing the sample population. Demographics are entirely missing. It is unclear how participants were grouped into ramen vs. no-ramen until page 5 halfway down the results section. It is also unclear what measures were gained from the blood sample. 

Confusing is further the addition of variables in the results that were not previously mentioned and seem to have no relevance, e.g. why is there a group of men mentioned (section 3.1 last sentence) and why was diastolic blood pressure suddenly mentioned (section 3.3 near end)? 

Similarly, the analyses are poorly defined. For example, how many correlational analyses were performed and what were the exact variables? It appears that there were hundreds of variables (different food items, nutrients, microbiota, microbial diversity, blood markers) and potentially all of them were included in individual correlations, however, this is unclear for all types of analyses.

There is also no clear hypothesis. Statements are ; "We investigated the relationship between habitual diet intake, gut microbial diversity, and blood biochemistry in 224 healthy Japanese ...." (abstract); "understanding and evaluating the effect pf food intake on human health and longevity remains a crucial task" (p 2 end of first paragraph); "results will provide insights into eating habits to improve health and the intestinal environment" (p 2, bottom). Due to the number of random results and the lack of a focused research question, the results lack meaning and interpretation. This is also reflected in the discrepancy between the title (only ramen mentioned), the abstract (lacking information about blood markers), and the rest of the paper (many other variables, including other foods, alcohol and nutrients). 

Each analysis section would benefit from a brief summary of the findings and directions of the effects. It is also critical to report non-significant findings.

While the grammar and spelling throughout the manuscript is overall good, it is often unclear which direction an effect goes. The paper would benefit from being more clear. For example, please define Shannon index and alpha diversity in the introduction, so that all readers understand; it is helpful to mention what a high vs. low value means. Please define all variables clearly in the methods and statistics and explain which analysis is executed on which variables exactly. Additionally, the first section in the discussion about ramen would be better in the introduction.

Please include a more detailed participant section in which the sample is characterized, including by group (ramen vs no-ramen). This could include age, BMI, education, blood pressure, blood measures and so on. It would be a helpful for the reader if it would also show between-group statistics on these measures.

Lastly, it would be helpful to have a non-expert read the manuscript in order to point out where other unclear sections are. Throughout the paper, it is often hard to follow the story line. The pieces do not always logically flow. 

I would mainly suggest to narrow down the research question (and define one actually) and remove a lot of the measures that are not relevant. This would make the paper stronger and the message clearer.

Author Response

My major concern is the large variety of variables and tests performed and the lack of clear description and definition of variables and exact models. For example, there is no adequate section or table characterizing the sample population. Demographics are entirely missing. It is unclear how participants were grouped into ramen vs. no-ramen until page 5 halfway down the results section. It is also unclear what measures were gained from the blood sample. 
> Thank you for the advice. The overall demographic information is expressed in Table 1 along with phenotype metadata information. This includes information on habitual diet intake, gut microbial diversity, and blood biochemistry covered in this study. Figures S2A and S2B describe how it was classified into the Ramen intake group and the no_intake group (L219-226). As you commented, we have prepared additional tables and explanations between the Ramen intake group and the no_intake group (Table S3, L226-229).

Confusing is further the addition of variables in the results that were not previously mentioned and seem to have no relevance, e.g. why is there a group of men mentioned (section 3.1 last sentence) and why was diastolic blood pressure suddenly mentioned (section 3.3 near end)? 
> Thank you for the comment. As you commented, we have eliminated the result of male which has caused confusion.

Similarly, the analyses are poorly defined. For example, how many correlational analyses were performed and what were the exact variables? It appears that there were hundreds of variables (different food items, nutrients, microbiota, microbial diversity, blood markers) and potentially all of them were included in individual correlations, however, this is unclear for all types of analyses.
> Thank you for the advice. As you commented, we changed the flow of results and added an explanation for the analysis.

There is also no clear hypothesis. Statements are ; "We investigated the relationship between habitual diet intake, gut microbial diversity, and blood biochemistry in 224 healthy Japanese ...." (abstract); "understanding and evaluating the effect pf food intake on human health and longevity remains a crucial task" (p 2 end of first paragraph); "results will provide insights into eating habits to improve health and the intestinal environment" (p 2, bottom). Due to the number of random results and the lack of a focused research question, the results lack meaning and interpretation. This is also reflected in the discrepancy between the title (only ramen mentioned), the abstract (lacking information about blood markers), and the rest of the paper (many other variables, including other foods, alcohol and nutrients). 
> Thank you for the advice. we added appropriate content to the introduction part (L100-117) and added appropriate effect directions and hypotheses to each session of the results.

Each analysis section would benefit from a brief summary of the findings and directions of the effects. It is also critical to report non-significant findings.
> As you commented, we added the benefit and direction of the previous results to the first of each session, and we also added the negative results (L248-251).

While the grammar and spelling throughout the manuscript is overall good, it is often unclear which direction an effect goes. The paper would benefit from being more clear. For example, please define Shannon index and alpha diversity in the introduction, so that all readers understand; it is helpful to mention what a high vs. low value means. Please define all variables clearly in the methods and statistics and explain which analysis is executed on which variables exactly. Additionally, the first section in the discussion about ramen would be better in the introduction.
> Thank you for the advice. We have added a brief description of the application according to your suggestion (L178-179). Information related to Ramen has also been moved to the Introduction part.

Please include a more detailed participant section in which the sample is characterized, including by group (ramen vs no-ramen). This could include age, BMI, education, blood pressure, blood measures and so on. It would be a helpful for the reader if it would also show between-group statistics on these measures.
> The relevant information was organized in Table S3 and the contents were added.

Lastly, it would be helpful to have a non-expert read the manuscript in order to point out where other unclear sections are. Throughout the paper, it is often hard to follow the story line. The pieces do not always logically flow. I would mainly suggest to narrow down the research question (and define one actually) and remove a lot of the measures that are not relevant. This would make the paper stronger and the message clearer.
> Thank you for the kind advice. We discussed with co-authors and revised the overall flow and content.

Reviewer 2 Report

Park et al attempted to investigate if the gut microbial composition is associated with ramen intake. To achieve this, they leveraged a cohort of 224 healthy Japanese adult females aged 27–80 years in a cross-sectional study. They used 16S sequencing of their fecal samples to determine their microbiota compositions, finding the Shannon index negatively correlates with the ramen intake. In the ramen-intake group, Phascolarctobacterium, Dorea, Providencia, Eubacterium eligens, and Anaerostipes have low abundances than those in the non-intake group. Besides, they assessed their habitual diets using a brief self-administered diet history questionnaire (BDHQ). In addition to the correlation between the relative abundance of each species with the ramen intake, they also investigated the correlation between other nutrient intakes and the ramen intake. They found that the ramen-intake group had a significantly lower intake of vitamins, minerals, and dietary fiber than the non-intake group.

Major issues:

1.     Throughout the paper, the authors emphasized the connection between ramen intake and other factors such as microbial abundance, community diversity, and intakes of other nutrients. However, in the introduction, they did not mention any background information about ramen nor give a motivation for why they focused on ramen in this study. I suggest they add more background and motivation so that readers can better appreciate their results.

2.     I think a more detailed discussion of how the ramen intake influences the gut microbes and how gut microbes impact the metabolism and blood metabolites is lacking. Does ramen intake change the gut microbiome then the gut microbiome changes the metabolism (Tong Wang et al., PloS Computational Biology 2019)?

Minor comments:

1.     Line 158-159: “. ramen intake showed…” --> “. Ramen intake showed…”

2.     Line 198: “nutrients intake, and gut bacteria.” -> “nutrient intake, and gut bacteria.”

3.     Line 205: “ramen are generally regarded…” -> “ramen is generally regarded…”

4.     Line 256: “Nutrient intake” -> “nutrient intake”

Author Response

Major issues:

1. Throughout the paper, the authors emphasized the connection between ramen intake and other factors such as microbial abundance, community diversity, and intakes of other nutrients. However, in the introduction, they did not mention any background information about ramen nor give a motivation for why they focused on ramen in this study. I suggest they add more background and motivation so that readers can better appreciate their results.

> Thank you for the advice. We moved the background information about Ramen from the discussion part to the introduction part and added research background information related to Ramen (L104-114)

2. I think a more detailed discussion of how the ramen intake influences the gut microbes and how gut microbes impact the metabolism and blood metabolites is lacking. Does ramen intake change the gut microbiome then the gut microbiome changes the metabolism (Tong Wang et al., PloS Computational Biology 2019)?

> Thank you for the advice. As you commented, we added a discussion of the relationship between Ramen intake and gut bacteria, and the resulting effects on metabolism and blood metabolism (L348-362)

Minor comments:

  1. Line 158-159: “. ramen intake showed…” --> “. Ramen intake showed…”
  2. Line 198: “nutrients intake, and gut bacteria.” -> “nutrient intake, and gut bacteria.”
  3. Line 205: “ramen are generally regarded…” -> “ramen is generally regarded…”
  4. Line 256: “Nutrient intake” -> “nutrient intake”

> Thank you for the kind advice. As you commented, we revised the contents.

Round 2

Reviewer 2 Report

The authors answered all my questions satisfactorily. I have no further questions.